# Effects of Sedentary Behavior Interventions on Mental Well-Being and Work Performance While Working from Home during the COVID-19 Pandemic: A Pilot Randomized Controlled Trial

**DOI:** 10.3390/ijerph19116401

**Published:** 2022-05-24

**Authors:** Grace E. Falk, Emily L. Mailey, Hayrettin Okut, Sara K. Rosenkranz, Richard R. Rosenkranz, Justin L. Montney, Elizabeth Ablah

**Affiliations:** 1School of Medicine, University of Kansas School of Medicine, Wichita, KS 67214, USA; 2Department of Kinesiology, College of Health and Human Sciences, Kansas State University, Manhattan, KS 66506, USA; emailey@ksu.edu (E.L.M.); jlmontney@ksu.edu (J.L.M.); 3Department of Population Health, University of Kansas School of Medicine, Wichita, KS 67214, USA; hokut@kumc.edu (H.O.); eablah@kumc.edu (E.A.); 4Department of Food, Nutrition, Dietetics and Health, College of Health and Human Sciences, Kansas State University, Manhattan, KS 66506, USA; sararose@ksu.edu (S.K.R.); ricardo@ksu.edu (R.R.R.)

**Keywords:** mental well-being, work performance, sedentary behavior, mood, energy

## Abstract

Sedentary behavior negatively impacts mental health, which can decrease employee productivity. Employee mental well-being and work performance may improve with sedentary reduction interventions, especially strategies that include environmental workplace modifications and behavior-changing strategies. However, such interventions have not been examined among employees working remotely during the COVID-19 pandemic. As part of the Stand Up Kansas program, 95 sedentary university employees working from home were randomized into one of four intervention arms: height-adjustable desk provision (Desk Only), online sedentary behavior modification program (Program Only), Desk + Program, or Control. The outcomes were measured at a baseline (November 2020) and following the 12-week intervention (February 2021). Employees reported mood (positive and negative affect), stress, fatigue (duration, interference with activities and severity) and work performance (irritability, focus, work satisfaction, non-work satisfaction and productivity) were measured using established self-report instruments. The effect sizes, by comparing the Control arm to the Desk + Program arm, revealed large improvements in mood (positive affect, *d* = 1.106). Moderate improvements were also seen in fatigue (duration, *d* = −0.533, and interference with activities, *d* = −0.648) and several aspects of work performance (focus, *d* = 0.702, work satisfaction, *d* = 0.751, and productivity, *d* = 0.572). Moderate effect sizes were also seen for positive affect (*d* = 0.566) and fatigue severity (*d* = 0.577) among the Program Only arm, whereas no noteworthy effect sizes were observed among the Desk Only arm. Combining an online sedentary behavior modification program with height-adjustable desk provisions appeared to positively affect mental well-being and work performance among remote employees.

## 1. Introduction

High levels of sedentary behavior, defined as “any waking behavior characterized by an energy expenditure ≤1.5 metabolic equivalents (METs), while in a sitting, reclining, or lying posture [1]”, are associated with a variety of adverse physical health outcomes among adults, including increased incidence of cardiovascular disease, type two diabetes mellitus, cancer and all-cause mortality [2]. Mental health may also be negatively impacted by sedentary behavior, as more sedentary time has been associated with a higher risk of experiencing symptoms of depression or anxiety [3,4]. Additionally, work-related sedentary behavior and poor mental health status have both independently been associated with lower productivity levels at work [5,6]. Work-related sedentary behavior is especially concerning for the U.S. workforce, as approximately 80% of employees work in sedentary jobs [7].

Given the negative impacts of sedentary behavior on employee health [2,3,4], environmental and behavioral interventions have been implemented to combat sedentary behavior in the workplace. Environmental interventions primarily alter the physical space where employees work to promote less sedentary behavior, whereas behavioral interventions aim to alter employee physical activity and/or to modify sedentary behavior patterns using strategies such as individual motivational techniques, changes in social support at work or computer or smart phone prompts to stand or move. Previous behavioral interventions have decreased sedentary behavior among employees by an average of approximately 15 to 45 min daily [8,9,10,11]. One parallel-group randomized trial reported that a behavioral program containing educational sessions, electronic reminders, self-monitoring and accountability strategies to promote frequent, short breaks from sitting was successful in improving female employees’ reported mood, fatigue and productivity without any environmental interventions [12]. Overall, however, little is known about the relationship between sedentary behavioral interventions that do not alter the physical work environment and employee-reported mental well-being or productivity. However, the impacts of behavioral interventions, combined with environmental interventions, have been studied more extensively, for example, by the *Stand More AT (SMArT) Work* and *Stand Up Victoria* trials [13,14]. These studies indicated that employee fatigue levels, work satisfaction and work performance improved after implementing combined behavioral and environmental sedentary behavior interventions [13,14]. Despite these data, concerns about productivity losses are frequently cited by employers as barriers to implementing sedentary behavior interventions [15].

The primary environmental intervention used to reduce sedentary behavior in the workplace has been the provision of height-adjustable desks, which allow employees to alternate between sitting and standing postures while working [9]. This intervention decreases sedentary time by an average of 30 to 120 min daily [16], but little is known about the effects of the provision and the use of a height-adjustable desk on mental well-being indicators, such as employee reported mood, stress and fatigue, or work performance indicators, such as employee focus, work satisfaction and productivity. Two randomized trials have indicated that implementing height-adjustable desks is associated with improved employee mood, fatigue, well-being, energy levels, and/or task engagement [17,18]; however, another randomized crossover trial indicated that height-adjustable desk implementation does not impact these outcomes [19]. Most studies have suggested that the use of height-adjustable desks alone is not associated with a sustained change in employee productivity [17,18,20,21,22].

Recently, the unforeseen workplace challenges that have accompanied the COVID-19 pandemic have prompted additional concerns about workforce sedentary behavior, mental well-being and job performance. Socially distanced employees have reported increased sedentary time, more negative mood, increased stress levels, decreased energy levels, poorer sleep quality, decreased concentration, decreased satisfaction with work and decreased productivity compared to prior to the pandemic [23,24,25]. These negative impacts associated with working from home during the COVID-19 pandemic are especially relevant given that approximately 70% of employees who previously worked at an office desk have worked from home at some point during the pandemic [25]. Therefore, the purpose of this pilot study was to determine whether a behavioral intervention program and/or environmental intervention is associated with improvements in sedentary employees’ mental well-being, including mood, stress levels and fatigue, without negatively impacting work performance, including employee irritability, focus, work satisfaction, non-work satisfaction, and productivity, while working from home during the COVID-19 pandemic.

## 2. Materials and Methods

### 2.1. Study Design and Procedures

The design of this pilot randomized controlled trial was pre-registered at clinicaltrials.gov (NCT04641689) and has been described by Mailey et al. [26]. The primary aim of the previously reported trial was to evaluate whether a sedentary behavior reduction program, height-adjustable desk provision, or a combination of these two interventions impacted occupational sedentary behavior among employees working from home during the COVID-19 pandemic [26]. The effects of the interventions on sedentary behavior and secondary cardiometabolic health outcomes have been previously published [26]. The current investigation reports secondary outcomes regarding changes in employees’ mental well-being, assessed via reported mood, stress and fatigue, as well as regarding changes in employee work performance, including irritability, focus, work satisfaction, non-work satisfaction and productivity, in each intervention arm. Given that these secondary outcomes could not be sufficiently powered in the original trial, the current study is considered a pilot study [26].

Briefly, participants who met the inclusion criteria were sent online questionnaires via a Qualtrics survey that assessed baseline mental well-being and work performance. After obtaining this information and stratifying by sex and BMI, employees were randomized into one of four intervention arms: Desk Only, Program Only, Desk + Program, or Control [26]. All outcomes were assessed pre- and post-intervention, which lasted 12 weeks, from November 2020 to February 2021.

### 2.2. Intervention

The protocols used for the height-adjustable desk and for the behavioral intervention program have been previously described [26]. Briefly, the Varidesk ProPlus 36 model height-adjustable desk and Varidesk anti-fatigue standing mat were provided to those in the Desk Only and Desk + Program trial arms. Participants were asked to document that their desk was set up appropriately by sending photos of the setup to the research team and received general advice to use the height-adjustable desk to reduce sitting time while working. Weekly online behavioral modules, informed by Social Cognitive Theory and designed to promote feasible, progressive changes in sedentary behavior and physical activity [27,28], were provided to those in the Program Only and Desk + Program trial arms. This program incorporated many behavior-changing techniques, including goal setting, problem solving, self-monitoring, social support and habit formation [26]. Control arm participants did not receive access to the height-adjustable desk nor to the behavioral program until after all post-intervention data had been collected.

### 2.3. Participants

The participants were university employees who volunteered for the study after recruitment via university announcements [26]. The participants were aware that they could receive a free height-adjustable desk by participating in the trial. The inclusion criteria for the study were: employees had to work from home at least 80% of the time, work 30 or more hours per week and spend at least 75% of their working time seated [26]. The Occupational Sitting and Physical Activity Questionnaire [29] was used to assess whether employees met the requirements for working time spent seated. Participants were excluded if they had a medical condition that would limit their ability to stand or if they already had a height-adjustable desk. The sample size was capped at 100 participants, based on the number of available desks and mats [26].

### 2.4. Instruments

Demographic information, including participant age, sex and body mass index (BMI; calculated via objectively measured height and weight) was collected at the baseline [26]. Four self-report instruments were used to evaluate the outcomes of interest in this study: the Positive and Negative Affect Schedule (PANAS), the Perceived Stress Scale (PSS), the Fatigue Symptom Inventory (FSI) and the Health and Work Questionnaire (HWQ) [30,31,32,33]. All instruments were administered online pre- and post-intervention. The post-intervention survey also included several open-ended questions to qualitatively assess the intervention effects among participants who received the height-adjustable desk and/or the behavioral program.

To assess mood, which is primarily influenced by positive and negative affect [34], the PANAS was used. This instrument has shown adequate psychometric properties in measuring positive and negative affect among the general population [35]. PANAS scores indicate associations between higher negative affect scores and depression and anxiety symptoms, and associations between lower positive affect scores and depression symptoms [30,35]. Participants reported the degree to which they experienced different positive (e.g., excited or inspired) and negative (e.g., upset or irritable) feeling states during the past week on a scale ranging from 1 (not at all) to 5 (extremely). The summed scores range from 10 to 50 for the positive and negative affect subscales, with higher scores reflecting higher positive affect and negative affect, respectively [30].

To assess stress, the four-question PSS was used, which has similar reliability levels but better internal validity for reporting stress levels than the ten-question PSS [36,37]. Additionally, it was advantageous to use the four-question PSS in the current study because this scale is shorter and thus easier for participants to complete than the ten-question scale [38]. Respondent scores for the abbreviated PSS are summed to yield a total score ranging from 0 to 16, with higher scores reflecting higher stress levels [31].

The FSI was used to assess employee fatigue. This scale consists of 13 questions that assess fatigue duration, interference with daily activities and severity level, with possible responses ranging from 0 to 10. Higher FSI scores are associated with more severe degrees of fatigue duration, interference with activities and severity [32,39]. The results of each subscale (2 questions for fatigue duration, 7 for fatigue interference and 4 for fatigue severity) are averaged to yield a total score.

To evaluate employee work performance, the HWQ was used [33]. This survey measures several constructs related to job performance: irritability, focus, work satisfaction, non-work satisfaction and productivity [33]. Office employees’ self-reported scores from the HWQ adequately measure productivity, as reported by employee supervisors [40]. For each subscale, respondents’ results were averaged to produce a score from 1 to 10, with higher scores reflecting higher levels of irritability, focus, work satisfaction, non-work satisfaction and productivity [33].

Qualitative questions developed by the study team provided additional insight into the intervention effects on study participants. Participants responded to the following questions on the post-intervention survey: please describe any positive changes (e.g., in physical or mental health, work performance, etc.) you have noticed since using the desk (Desk Only and Desk + Program arms); and please describe any positive changes you have made since starting this program (Program Only and Desk + Program arms).

### 2.5. Statistical Analysis

Frequencies, proportions, means and standard deviations were calculated for categorical and continuous variables. Prior to the main analyses, Shapiro–Wilks tests were conducted to test the normal distribution assumption for the outcome variables. Change scores were calculated for all outcomes by subtracting the baseline value from the post-intervention value. Effect sizes (Cohen’s *d*) were calculated to reflect the magnitude of variable changes from pre- to post-intervention between the Desk Only, Program Only and Desk + Program intervention arms versus the Control arm.

For the open-ended questions, one investigator read through the full list of responses and created a preliminary list of themes for each intervention (desk and program). A second investigator then reviewed the themes, and some were discarded or combined after checking them against the full dataset. The first investigator then coded the full list of responses, with some responses being assigned to more than one theme if appropriate. The second investigator checked all codes, and a small number of codes were revised following discussion. Finally, frequencies were calculated by dividing the number of times each theme was mentioned by the total number of participants who responded to each question.

## 3. Results

A full CONSORT diagram from the initial pilot study has been previously published [26] and is available as Appendix A. Briefly, of the 167 eligible volunteers, 95 completed all baseline measures and were randomized into the four interventions. Of these initial participants, 6 were lost to follow-up. The reasons for loss to follow-up included inability to reach participants at the end of the study, leaving employment at the university, an injury unrelated to the study and reported personal life stressors. Demographic variables including age, sex and BMI are reported in Table 1. There were no differences in the distribution of age, sex or BMI between the intervention arms. Sedentary behavior was previously reported, and baseline sedentary time is also included in Table 1 [26].

Descriptive statistics for all mental well-being and work performance scores are reported by intervention arm in Table 2. Among the three outcomes demonstrating significant pre- to post-intervention changes for the Desk + Program arm compared to the Control, the Cohen’s *d* effect size was large for positive affect (*d* = 1.106) and moderate for focus (*d* = 0.702) and work satisfaction (*d* = 0.751). Although not statistically significant, moderate effect sizes were also noted for improvements in productivity (*d* = 0.572), non-work satisfaction (*d* = 0.603), fatigue duration (*d* = −0.533) and fatigue interference with activities (*d* = −0.648) in the Desk + Program arm compared to the Control. Among the Program Only arm compared to the Control, fatigue worsened with a moderate effect size (fatigue severity, *d* = 0.577). Conversely, positive affect improved, with a moderate effect size between the Program Only and Control arms (*d* = 0.566). Regarding mood in the Desk Only arm, no effect was noted for positive affect change, whereas negative affect worsened compared to the Control (*d* = 0.434, Table 2).

Qualitative themes reflecting positive changes that participants noticed after utilizing the height-adjustable desk and/or behavioral intervention are reported in Table 3 and Table 4. Among the participants utilizing the height-adjustable desk, the noted response themes included improvements in work performance (49% of responses), improvements in energy levels (39% of responses) and improvements in mental well-being (17% of responses, Table 3). One participant noted, “I have more energy. I feel more alert and more motivated to do work,” and another said they were “not as fatigued during the workday which helps with work performance, mental health, etc.”

For participants utilizing the behavioral program, notable themes included increased physical activity during the workday (45% of responses), an increased awareness of healthy habits (36% of responses) and improvements in work performance (14% of responses, Table 4). One respondent noted that, after completing the behavioral program, they were “more aware of how much I am sitting during the day and try to get up and move more often.” Another reported, “I’ve learned to associate standing with other tasks and activities I have to do throughout the day and use my breaks to incorporate these tasks and get them done periodically throughout my day. This has helped me break up my workday and take needed breaks from sitting while still getting necessary tasks accomplished.” Other participants reported that they were “Eating better. Less fatigued after zooming” and had “better focus”.

## 4. Discussion

The primary effects of these environmental and behavioral interventions on sedentary behavior have previously been reported, indicating that both environmental and behavioral programs are effective at reducing the number of minutes that remote employees spend sitting during the day (*d* = −0.98 and −1.13, respectively), and the combination of an environmental and behavioral program appears most effective at reducing sedentary behavior (*d* = −1.84) [26]. The current pilot study reports the effects of these interventions on secondary outcomes related to employee-reported mental well-being and job performance while working remotely during the COVID-19 pandemic. When compared to having no intervention, the combined provision of a height-adjustable desk with a behavioral program to reduce sitting in a work-from-home environment produced large improvements in employee mental well-being (mood, specifically positive affect). Moderate improvements were seen in fatigue and work performance (focus, work satisfaction, non-work satisfaction and productivity). The pilot nature of this study makes it inappropriate to determine the statistical significance of the results; however, the effect sizes of changes in positive affect, fatigue, focus, and work satisfaction, are still noteworthy and encouraging because low mood, poor focus and high fatigue levels are associated with decreased employee productivity, whereas greater work satisfaction is associated with increased productivity [33,41,42,43]. The current study’s findings are similar to those of previous randomized controlled trials, such as *Stand More AT (SMArT) Work* and *Stand Up Victoria* [13,14]. These studies indicated that using a combination of environmental changes (height-adjustable desks) and behavioral interventions (including management buy-in, coworker engagement, educational sessions, individual goal setting and motivation, individual health coaching sessions and prompts to stand) were associated with reducing sedentary behavior and improving work performance [13,14].

However, the current study differs from previous trials in several ways. Firstly, the prior trials evaluated the combined effects of environmental and behavioral interventions [13,14], whereas the current study simultaneously analyzes the effects of a height-adjustable desk only, a behavioral intervention program only or the combination of these two interventions compared to having no intervention on employees’ mental well-being and work performance. Relative to the Control, more mental well-being and work performance outcomes had moderate to large improvements for the Desk + Program arm (positive affect, fatigue, focus, work satisfaction, non-work satisfaction and productivity) than in the Program Only arm (positive affect and fatigue) and Desk Only arm (none). Furthermore, although mood (positive affect) had moderate improvement in both the Program Only and Desk + Program arms, the effect size was large for Desk + Program but was moderate for Program Only. Overall, these findings suggest that concurrently tackling sedentary behavior using a multi-component approach (environmental and behavioral) may be needed to produce greater degrees of change in employees’ mental well-being and work performance. Additionally, given that productivity was not associated with negative effect sizes for the intervention arms relative to the Control, these findings suggest that this study’s environmental and behavioral interventions to reduce sitting did not negatively impact employee productivity.

The current study is also unique from previous trials because this study was conducted in a work-from-home environment during the COVID-19 pandemic, whereas previous trials have been conducted in traditional, office-based settings [13,14]. Before the onset of the COVID-19 pandemic, research on remote work suggested that distanced employees can face challenges that are distinct from traditional, in-person office employees, especially social isolation and increased stress from enhanced employee autonomy [44,45]. Although these challenges can be reduced or eliminated with appropriate organizational and managerial support [44], among the current study population, the shift to remote work meant that participants were working in an environment with distinct characteristics that previous, office-based, sedentary behavior reduction interventions have not considered. This difference is of particular interest when one considers the performance of the behavioral program implemented in the current study.

The behavioral program used in the current study relied on multiple behavior-changing techniques, including the provision and promotion of social support to reduce sitting and the promotion of small increases in physical activity [26]. Social support among coworkers is commonly included in office-based, sedentary behavior reduction programs [13,14,46]; however, the social support component in the current study had to be conducted entirely virtually, through discussion boards and video calls [26]. This support may have played a role in enhancing mood during a time of high social isolation among participants who received the behavioral program, as participants in both the Program Only and Desk + Program arms reported moderate to large improvements in positive affect, which is promising for future, fully powered studies. These findings are further supported by participant comments that highlighted improvements in attitude and mood after participating in the intervention.

Physical activity changes among employees using the behavioral program are an important consideration when analyzing the outcomes of the current study because the implementation of social-distancing protocols in response to the COVID-19 pandemic likely created additional barriers to changing the physical activity levels for the current study population [47]. The behavioral program may have helped participants develop creative strategies for overcoming these barriers and integrating small amounts of physical activity throughout the day. Indeed, the qualitative analysis indicated that 45% of participants using the behavioral program reported increased physical activity during the workday, and 36% reported that they increased their amount of exercise outside work. Given that prior research has indicated that increased physical activity is associated with improvements in indicators of mental well-being, such as mood and stress [48,49], the added emphasis on physical activity in the behavioral program may have contributed to improvements in mood among the Program Only and Desk + Program arms compared to the Control. These findings are similar to a previous study regarding the impacts of a sedentary behavior reduction program on female employees’ reported mood [12], despite the current study being conducted in a socially distanced format.

This is also the first study to evaluate the effects of height-adjustable desk provisions on remote-employee-reported mental well-being and work performance during the COVID-19 pandemic. Employee workstation design, which includes the desk, is important to consider when evaluating remote work outcomes, as having an inadequate home workstation setup has been associated with increased employee stress and decreased productivity [50]. At the end of the current study, participants using only a height-adjustable desk to reduce sitting at work did not report any moderate or large changes in mental well-being or work performance compared to the Control. These findings are consistent with prior research obtained from employees using height-adjustable desks in traditional office settings [17,19,20,21]. Although height-adjustable desk provisions alone while working from home did not improve employee mental well-being or work performance in a non-powered statistical analysis, the qualitative analyses indicated that some study participants did report benefits, such as increased energy and focus. Furthermore, productivity did not appear to worsen for participants using the height-adjustable desks. These findings are particularly relevant in the context of the COVID-19 pandemic, where challenges associated with socially distanced work have been associated with decreased employee productivity [45,51].

Prior to the pandemic, studies indicated that, with proper organizational support, employees working remotely can be equally, if not, more productive than when working in the office [44,52]. However, the rapid transition to working remotely and social distancing that occurred during the pandemic brought many challenges to office-based employees, including transitions to remote work without proper organizational support, inadequate home work stations, multiple family members trying to work remotely in the same space, having to care for children while working and the lack of social interaction with coworkers and friends [45,50,53,54]. These challenges can increase employees’ feelings of stress and loneliness [45,50,53,54], which is concerning because employees working from home during the COVID-19 pandemic who report higher levels of stress and social isolation also report lower productivity levels [45,51]. Reported mental well-being and work performance may have been influenced by regional COVID-19 pandemic events during the study period, such as surges in COVID-19 cases, hospitalizations, deaths, the early introduction of the COVID-19 vaccine and the first winter holiday season during the pandemic [55]. However, these regional pandemic factors likely impacted all study participants, including those randomized to the Control arm. The lack of effect sizes noted in the Control group relative to the Desk + Program arm indicates that, in the context of the challenges associated with socially distanced work during the COVID-19 pandemic, the observed changes in employees’ mental well-being and work performance for the current pilot study warrant employer consideration. The combination of using a height-adjustable desk with the behavioral intervention program to reduce employee sitting resulted in improvements in employee mood, focus, work satisfaction, non-work satisfaction and productivity.

### Limitations

This study is limited primarily by the number of desks available and thus by its sample size; therefore, the study was not fully powered to detect statistically significant differences in these secondary outcomes. Qualitative data were added to supplement the limited statistical tests, but the sample size still limits the generalizability of this study’s findings. Additionally, because the participants were aware that they could receive a free height-adjustable desk if they volunteered for the study, the study findings may be influenced by a motivated population. Differences in participant engagement in interventions may have also influenced the results, as the previous primary outcomes study indicated that individuals who viewed the online behavioral program content more had a greater reduction in their sedentary time compared to participants who viewed the content less [26]. This study is also limited by its duration (three months). Developing and maintaining sedentary behavior reduction in the workplace may require longer intervention durations [22], and the extent to which improvements in mental well-being or work performance were sustained beyond the end of the intervention is unknown. Work performance and productivity measurements are further limited by the lack of a standardized measure of productivity in the academic literature. The measure used for this study is intended to promote a quantifiable assessment of an employee’s own performance (which is inherently subjective). Although this survey is a useful tool for employees to self-report work performance [33], further studies would likely benefit from alternative measures of work performance and productivity, such as supervisor evaluations. Additional research is necessary to evaluate the statistical significance of the current pilot study’s results. In particular, blinded studies with larger sample sizes and longer study durations are needed to allow for the determination of statistical significance and to perform hypothesis testing.

## 5. Conclusions

This pilot study evaluates the benefits of using environmental and behavioral interventions to reduce sedentary behavior and to improve remote employees’ mental well-being and work performance. This study is the first to analyze how each of the three intervention types (environmental only, behavioral only or environmental with behavioral) impacts employee-reported mood, stress, fatigue and work performance in a COVID-19 work-from-home environment. The results indicate that utilizing a combination of environmental and behavioral program interventions generates moderate to large improvements in mental well-being and work performance, such as in mood, fatigue, focus, work satisfaction, non-work satisfaction and productivity.

## Figures and Tables

**Table 1 ijerph-19-06401-t001:** Demographics of Study Population at Pre-Intervention.

Variable	Desk Only (*n* = 24)	Program Only (*n* = 21)	Desk + Program (*n* = 21)	Control (*n* = 23)
Mean (SD) Age (years)	46.5 (10.03)	43.1 (12.32)	42.0 (9.36)	45.2 (10.79)
Number (%) of Female Participants	19 (79.2%)	16 (76.2%)	16 (76.2%)	19 (82.6%)
Number (%) of Participants of Normal BMI	6 (25.0%)	6 (28.6%)	5 (23.8%)	9 (39.1%)
Number (%) of Participants of Overweight BMI	10 (41.7%)	5 (23.8%)	8 (38.1%)	4 (17.4%)
Number (%) of Participants of Obese BMI	8 (33.3%)	10 (47.6%)	8 (38.1%)	10 (43.5%)
Mean (SD) Participant BMI	29.45 (4.83)	29.81 (6.38)	29.12 (5.57)	29.34 (6.56)
Mean (SD) Minutes Sitting per Workday	412.3 (94.2)	437.3 (81.3)	444.1 (63.82)	456.6 (55.8)

SD = standard deviation. A total of 95 participants were included in the study, and 6 were lost to follow-up (1 in the Desk Only arm, 2 in the Program Only arm, 2 in the Desk + Program arm and 1 in the Control arm). Only patients completing the follow-up surveys are included in this table. The full CONSORT diagram is available in the Appendix A section.

**Table 2 ijerph-19-06401-t002:** Mean Change in Outcomes with Effect Sizes.

-	Desk Only (*n* = 24)				Program Only (*n* =21)				Desk + Program (*n* = 21)				Control (*n* = 23)		
	Pre-Intervention Mean (SD)	Post-Intervention Mean (SD)	Mean Change	Cohen’s *d* ^a^	Pre-Intervention Mean (SD)	Post-Intervention Mean (SD)	Mean Change	Cohen’s *d* ^a^	Pre-Intervention Mean (SD)	Post-Intervention Mean (SD)	Mean Change	Cohen’s *d* ^a^	Pre-Intervention Mean (SD)	Post-Intervention Mean (SD)	Mean Change
Positive Affect	31.9 (9.0)	34.7 (9.7)	2.74	0.135	32.0 (8.4)	36.8 (7.1)	4.95	**0.566**	27.7 (6.4)	36.6 (7.0)	9.35	**1.106**	33.6 (6.6)	35.7 (8.1)	1.91
Negative Affect	17.3 (4.4)	18.1 (6.2)	0.74	0.434	20.6 (6.4)	17.9 (5.5)	−2.38	−0.124	19.3 (6.8)	16.8 (5.1)	−3.30	−0.266	19.2 (7.1)	17.7 (5.4)	−1.74
Stress	9.2 (2.7)	8.8 (2.3)	−0.48	0.156	9.7 (2.5)	8.4 (3.0)	−1.29	−0.177	9.4 (2.9)	8.3 (2.6)	−1.30	−0.147	8.4 (2.7)	7.6 (2.6)	−0.87
Fatigue Duration	4.2 (1.6)	4.0 (1.6)	−0.17	0.134	4.1 (1.8)	4.0 (1.9)	−0.12	0.182	5.3 (1.7)	4.0 (2.4)	−1.55	**−0.533**	4.5 (2.0)	4.0 (2.0)	−0.43
Fatigue Interference	2.4 (1.7)	2.0 (1.8)	−0.37	−0.076	2.7 (2.0)	1.7 (1.7)	−0.92	−0.484	2.9 (2.3)	1.9 (1.9)	−1.41	**−0.648**	2.3 (1.8)	2.1 (1.9)	−0.25
Fatigue Severity	3.6 (2.0)	3.5 (1.7)	−0.10	0.228	3.2 (1.6)	3.5 (2.2)	0.43	**0.577**	3.8 (1.7)	2.9 (2.3)	−0.90	−0.191	4.1 (2.0)	3.5 (2.3)	−0.51
Irritability	3.2 (1.8)	2.8 (1.6)	−0.22	−0.081	3.2 (1.9)	2.7 (1.6)	−0.62	−0.295	3.3 (1.9)	3.2 (1.5)	−0.35	−0.142	3.1 (1.1)	3.0 (1.9)	−0.08
Focus	6.9 (1.9)	7.3 (1.9)	0.15	0.069	7.0 (2.0)	7.5 (1.8)	0.40	0.201	6.5 (1.9)	7.5 (1.6)	1.26	**0.702**	7.1 (1.9)	7.0 (2.1)	0.02
Work Satisfaction	6.8 (1.5)	7.1 (1.5)	0.28	0.187	7.0 (1.8)	6.9 (1.7)	0.07	0.058	6.3 (1.7)	7.4 (1.5)	1.24	**0.751**	7.3 (1.5)	7.3 (1.6)	−0.02
Non-work Satisfaction	8.4 (1.5)	8.1 (1.6)	−0.26	−0.340	7.2 (1.6)	7.6 (1.7)	0.40	0.085	6.9 (2.1)	8.1 (1.8)	1.25	**0.603**	7.9 (1.7)	8.4 (1.3)	0.28
Productivity	7.2 (1.4)	7.4 (1.1)	0.15	0.090	7.2 (1.3)	7.4 (1.0)	0.36	0.257	7.0 (1.2)	7.6 (1.4)	0.70	**0.572**	7.4 (1.2)	7.5 (1.4)	0.05

*n* reflects the number of participants completing the post-intervention surveys in each intervention arm. SD = standard deviation. Possible ranges for each outcome variable include mood (positive and negative affect, 10 to 50), fatigue (duration, interference with activities and severity, 0 to 10) and work performance (irritability, focus, work satisfaction, non-work satisfaction and productivity, 0 to 10). ^a^ = effect size relative to Control. Bolded values indicate moderate and large effect sizes relative to Control.

**Table 3 ijerph-19-06401-t003:** Qualitative Analysis of Height-Adjustable Desk Benefits on Employee Mental Well-being and Work Performance.

Theme	% of Responses Including This Theme
Improved work performance (more focused, etc.)	49%
Improved energy	39%
Increased activity levels	32%
Less pain/stiffness/soreness	32%
Physiological health improvement	22%
Improved mental well-being (improved mood, etc.)	17%
Improvements in non-physical activity health behaviors (such as sleep)	5%

A total of 41 respondents from the Desk Only and Desk + Program arms provided qualitative feedback about the benefits they noticed after using a height-adjustable desk.

**Table 4 ijerph-19-06401-t004:** Qualitative Analysis of Behavioral Program Benefits on Employee Mental Well-being and Work Performance.

Theme	% of Responses Including This Theme
Increased physical activity during desk work	45%
Increased amount of exercise outside of work	36%
Increased awareness of healthy habits	36%
Taking more breaks	24%
Improved diet	21%
Increased work performance (focus, productivity, etc.)	14%
Physiological health benefits	12%
Spending more time outside	12%
No benefit	2%

A total of 42 respondents from the Program Only and Desk + Program arms provided qualitative feedback about the benefits they noticed from the sedentary behavior reduction program.

## Data Availability

The data collected and analyzed for this study are available on request from the corresponding author. The data are not publicly available in order to protect participant privacy.

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
