# Peer review of "Effects of Sedentary Behavior Interventions on Mental Well-Being and Work Performance While Working from Home during the COVID-19 Pandemic: A Pilot Randomized Controlled Trial"

_ijerph, 2022, doi:10.3390/ijerph19116401_

Round 1

Reviewer 1 Report

The purpose of this study was to determine whether a behavioral intervention program and/or environmental intervention was associated with improvements in sedentary employees’ mental well-being, while working from home during the COVID-19 pandemic.

Inactivity and all its negative consequences are increasingly present in our society. This study approaches a way to influence the level of inactivity of the population, in one of the daily activities that generates the most inactivity in many inhabitants, the workplace. The research is well introduced and designed.

Here are my comments:

  • It is unclear how many participants were included in the study. In the methodology appears 100, in the results 95 and 6 excluded….
  • In table 2, the title is in bold and the footnote is spaced further than in table one. Unify formatting.
  • Line 297, are references 43-44 correctly placed?
  • In the discussion, there is some commentary but little depth on the effect the pandemic may have had on the psychological values reported by the subjects.
  • Missing information: “Data Availability Statement: The data collected and analyzed for this study are publicly available on publication at __. “

Author Response

Reviewer 1 responses:

  • It is unclear how many participants were included in the study. In the methodology appears 100, in the results 95 and 6 excluded….
    • Thank you for pointing out this discrepancy. Based on the number of available desks, the maximum number of participants was initially set at 100. However, only 95 participants completed all baseline assessments and were randomized. This has been clarified in the manuscript in lines 212 to 213. Additionally, reasons for participant loss to follow up from the final analysis have been added to the consort flow diagram and the manuscript: lines 213-216.
  • In table 2, the title is in bold and the footnote is spaced further than in table one. Unify formatting.
    • Thank you for pointing out this formatting difference. We have made edits so that the title formatting is consistent across the tables. The footnote formatting has been fixed on Tables 2, 3, and 4 as well. Please note that the content of Table 2 was also edited based on another reviewer’s feedback.
  • Line 297, are references 43-44 correctly placed?
    • Yes, these references are placed in the correct place.
  • In the discussion, there is some commentary but little depth on the effect the pandemic may have had on the psychological values reported by the subjects.
    • Thank you for your feedback. A brief discussion on pandemic events that might have impacted the study results has been added to the discussion section, lines 381-388.
  • Missing information: “Data Availability Statement: The data collected and analyzed for this study are publicly available on publication at __.
    • Thank you for pointing out this oversight. A data availability statement has been added to lines 449-451.

Reviewer 2 Report

This is a well written secondary article to the main paper. I attach a word doc with my minor suggestions for improvement.

Author Response

Reviewer 2 responses:

  • ABSTRACT Tweak title to Sedentary Behavior Reduction Intervention? Not interventionS
    • Thank you for your feedback. Given that there were three total interventions (a height adjustable desk only, a behavioral program only, and both the desk and program), the authors feel that it is more appropriate to leave the title in the plural form.
  • Combining an online sedentary behavior modification program with height-adjustable desk provision appeared to positively affect mental wellbeing and work performance – need to explain earlier in abstract that ‘program’ = ‘online sedentary behavior modification program’ and ‘desk’ = ‘heightadjustable desk provision’.
    • Thank you for your feedback. We have updated the abstract to better illustrate these points on lines 21-22 of page 1.
  • Spell out that it is only the combined intervention that showed benefit for these secondary outcomes?
    • Thank you for your feedback. We have added language to highlight this in lines 31-33.
  • Supplementary figure: n=167 minus 71 = 96 not n=95 – what happened to the missing person?
    • Thank you for bringing this to our attention. The sum of those eligible and ineligible reflects the total number of participants assessed for eligibility (238). Those who were eligible but not randomized did not complete all baseline assessments. It is purely coincidental that subtracting the 71 ineligible from the 167 eligible resulted in a value close to 95. This has been updated in the CONSORT documentation and in the manuscript, lines 212-216.
  • Low dropout – dropouts were excluded from analysis – could study authors have used baseline data for dropouts? Intention to Treat analysis or give reason for not doing this? What were the reasons for dropout?
    • Thank you for your feedback. The reasons for participant dropout have been added to the consort diagram and the manuscript on lines 213-216. Given the pilot nature of this study, the authors feel that an intention to treat analysis is not necessary. Please see: https://www.nccih.nih.gov/grants/pilot-studies-common-uses-and-misuses.
  • Stand Up Kansas - Mention name of trial in abstract so reviewers can identify it when searching databases and can link with other papers
    • Thank you for your feedback. This has been added to line 20.
  • Materials and Method: Please be explicit that this is a paper that reports secondary outcomes and the primary outcome and other health-related secondary outcomes are in the main paper: Reducing Occupational Sitting While Working From Home: Individual and Combined Effects of a Height-Adjustable Desk and an Online Behavioral Intervention | Request PDF (researchgate.net)
    • Thank you for your feedback. Language highlighting that this paper reports secondary outcomes from the trial can be found in lines 103-108.
  • Table 1: Please add baseline fitness (could be sedentary for work hours but participants could vary in physical activity outside of work hours) and baseline sedentary time?
    • Thank you for the feedback. Baseline fitness data were not collected and thus cannot be reported. Baseline sedentary time and for each group was previously reported in the primary outcomes paper. Table 1 has been updated to now include these data. The mean BMI for each of the intervention arms has also been added to Table 1. Additionally, in accordance with another reviewer’s feedback, Table 1 has been edited to only report the demographic characteristics of those completing the study.
  • Table 2: put groups in same order as table 1 ie ‘control’ at end
    • Thank you for the feedback. Table 2 has been updated to reflect these suggestions.
  • Line 298 – is there a word missing?
    • Thank you for pointing this out. We have re-read this section and added the missing word.
  • Results: I want to know if the intervention did reduce sedentary behavior (even though this is reported elsewhere in the main paper).
    • Thank you for your feedback. Baseline minutes spent sitting during work has been added to Table 1. Additionally, a brief summary of the changes in sedentary behavior for the four intervention arms has been added to the discussion section, line 273-277.
  • Also I would like a mention of adherence/engagement to the interventions
    • Thank you for your feedback. A brief summary of adherence/engagement to the interventions has been added to the limitations section on lines 401-404.
  • Discussion – v good but word count could be reduced a bit, might be worth comparing with results from main paper? For example, reduction in sedentary behavior associated with improvements in some secondary outcomes such as mood and fatigue but not health outcomes as this would require more time?
    • Thank you for your feedback. We have added a brief discussion of sedentary behavior changes noted in the primary study to the current discussion, lines 273-277.
  • Notes: Motivated sample because not blinded and knew would all receive a free desk to take home – this could be further discussed within the discussion section. Could the desk provision provide other indirect benefits ie feel more valued as an employee?
    • Thank you for your feedback. Discussion of a motivated sample has been added to the limitation section of the discussion on lines 399-401.
  • Check Country specific context of COVID-19 during RCT delivery Nov 20 to Feb 21 – anything significant that might have affected intervention?
    • Thank you for your feedback. A brief discussion of regional pandemic events that might have impacted the results has been added to the discussion section: lines 381-388.
  • SCT is reported in trial registry data but not in paper?
    • Thank you for bringing this to our attention. We have added a reference to SCT to the description of the intervention, line 128.
  • I presume COI is declared (varidesk) and role of funder in the study
    • Yes, Varidesk’s contributions are listed in the conflict of interest section on line 452-454.

Reviewer 3 Report

This study is of interest because it aims to investigate the possible association between a behavioral intervention program and/or environmental impact with improved mental well-being in sedentary employees. This topic has become especially important during the COVID-19 pandemic and remote work for many employees.

However, while reading the manuscript, I had a number of comments that should be clarified and/or eliminated.

1. There are too many references in this manuscript to previous work by the authors in the study cohort (Reducing Occupational Sitting While Working From Home: Individual and Combined Effects of a Height-Adjustable Desk and an Online Behavioral Intervention; doi: 10.1097/JOM.0000000000002410). If in the material and methods section such references look justified and appropriate, then in the results section it already looks incorrect. For example, when interpreting Table 1, the authors indicate that the differences between the studied groups were statistically nonsignificant and provide a link to their previous publication. Nevertheless, the differences between the groups in terms of individual indicators look quite substantial (the number of patients with  Overweight BMI in the groups Desk Only and Control differs by more than 2 times - 44% and 16.67%). Therefore, I consider it necessary to present the statistical values ​​of the differences (p) between the groups in this table.

2. When reviewing the flow chart of the study, it can be noted that the number of individuals included in the study and the number of individuals included in the analysis differ (95 and 89, respectively). At the same time, Table 1 provides data on 95 examined persons, and Table 2 - on 89 examined persons. It would be more correct to present data in Table 1 only for those persons who are included in the analysis, that is, for 89 examined persons.

3. Table 2 shows data on the statistical significance of changes in the studied parameters in each of the groups compared with the control using Student's t-test. In fact, only pairwise comparisons were made in the presence of 4 groups. Apparently, this analysis is incorrect, since it does not take into account the effect of multiple comparisons in the presence of more than 2 studied groups.

4. Many questions are caused by the data presented in tables 3-4. First, why are the results of two groups combined in these tables (in table 3 - these are the groups Desk Only and Desk + Program, in table 4 - the groups Program Only and Desk + Program). For example, in Table 2, the authors quite successfully analyze the differences between the 4 groups. Secondly, the refusal of the authors of the statistical processing of the results presented in tables 3-4 is also incomprehensible. The authors' argument that this analysis is only a pilot study does not look convincing, since for the results presented in Table 2 this did not prevent them from conducting such an analysis.

Author Response

Reviewer 3 responses:

  • This study is of interest because it aims to investigate the possible association between a behavioral intervention program and/or environmental impact with improved mental well-being in sedentary employees. This topic has become especially important during the COVID-19 pandemic and remote work for many employees.
    • Thank you. We are excited to share this study.
  • There are too many references in this manuscript to previous work by the authors in the study cohort (Reducing Occupational Sitting While Working From Home: Individual and Combined Effects of a Height-Adjustable Desk and an Online Behavioral Intervention; doi: 10.1097/JOM.0000000000002410). If in the material and methods section such references look justified and appropriate, then in the results section it already looks incorrect. For example, when interpreting Table 1, the authors indicate that the differences between the studied groups were statistically nonsignificant and provide a link to their previous publication. Nevertheless, the differences between the groups in terms of individual indicators look quite substantial (the number of patients with  Overweight BMI in the groups Desk Only and Control differs by more than 2 times - 44% and 16.67%). Therefore, I consider it necessary to present the statistical values ​​of the differences (p) between the groups in this table.
    • Thank you for your feedback. We have added the mean BMI of the study groups to help clarify the lack of differences between the study groups in Table 1. Regarding baseline statistical comparisons of participant demographics in the four intervention arms, the authors do not feel that this is justified, please see the following articles:
      • de Boer MR, Waterlander WE, Kuijper LD, Steenhuis IH, Twisk JW. Testing for baseline differences in randomized controlled trials: an unhealthy research behavior that is hard to eradicate. Int J Behav Nutr Phys Act. 2015;12:4.
      • Harvey LA. Statistical testing for baseline differences between randomised groups is not meaningful. Spinal Cord. 2018;56(10):919
    • When reviewing the flow chart of the study, it can be noted that the number of individuals included in the study and the number of individuals included in the analysis differ (95 and 89, respectively). At the same time, Table 1 provides data on 95 examined persons, and Table 2 - on 89 examined persons. It would be more correct to present data in Table 1 only for those persons who are included in the analysis, that is, for 89 examined persons.
      • Thank you for your feedback. Table 1 has been updated to display the demographic characteristics only for those included in the final analysis.
    • Table 2 shows data on the statistical significance of changes in the studied parameters in each of the groups compared with the control using Student's t-test. In fact, only pairwise comparisons were made in the presence of 4 groups. Apparently, this analysis is incorrect, since it does not take into account the effect of multiple comparisons in the presence of more than 2 studied groups.
      • Thank you for your feedback. Based on the pilot nature of this study, we have removed the p values from Table 2 and are only reporting the effect sizes.
    • Many questions are caused by the data presented in tables 3-4. First, why are the results of two groups combined in these tables (in table 3 - these are the groups Desk Only and Desk + Program, in table 4 - the groups Program Only and Desk + Program). For example, in Table 2, the authors quite successfully analyze the differences between the 4 groups. Secondly, the refusal of the authors of the statistical processing of the results presented in tables 3-4 is also incomprehensible. The authors' argument that this analysis is only a pilot study does not look convincing, since for the results presented in Table 2 this did not prevent them from conducting such an analysis.
      • Thank you for your feedback. Based on the pilot nature of this study, we have removed the p values from Table 2 and are only reporting the effect sizes. Given the pilot nature of this study, the goal was not to determine statistical significance, but rather to provide a preliminary indication of the potential of these interventions to be carried out and have an effect on meaningful outcomes. Please see: https://www.nccih.nih.gov/grants/pilot-studies-common-uses-and-misuses. It is our hope that the findings of this pilot trial can inform future, powered trials of sedentary behavior intervention impacts on remote employee mental wellbeing and work performance. By this same logic, the qualitative component in Tables 3 and 4 is reported by intervention type, since we believe the participant experience with each of the independent intervention approaches will be most helpful for future intervention work.

Round 2

Reviewer 3 Report

The authors have done sufficient work to improve their manuscript. I have no more comments.